# Uniting against a common enemy: Perceived outgroup threat elicits ingroup cohesion in chimpanzees

**James Brooks**[1,2]*, **Ena Onishi**[1,2], **Isabelle R. Clark**[3], **Manuel Bohn**[4], **Shinya Yamamoto**[1,5]

**1** Wildlife Research Center, Kyoto University, Kyoto, Japan, **2** Kumamoto Sanctuary, Kyoto University, Kumamoto, Japan, **3** Department of Anthropology, University of Texas at Austin, Austin, Texas, United States of America, **4** Department of Comparative Cultural Psychology, Max Planck Institute for Evolutionary Anthropology, Leipzig, Germany, **5** Institute for Advanced Study, Kyoto University, Kyoto, Japan

* jamesgerardbrooks@gmail.com

**Data Availability Statement:** All relevant data are within the paper and its Supporting Information files.

## Abstract

Outgroup threat has been identified as an important driver of ingroup cohesion in humans, but the evolutionary origin of such a relationship is unclear. Chimpanzees (*Pan troglodytes*) in the wild are notably aggressive towards outgroup members but coordinate complex behaviors with many individuals in group hunting and border patrols. One hypothesis claims that these behaviors evolve alongside one another, where outgroup threat selects for ingroup cohesion and group coordination. To test this hypothesis, 5 groups of chimpanzees (N = 29 individuals) were observed after hearing either pant-hoots of unfamiliar wild chimpanzees or control crow vocalizations both in their typical daily environment and in a context of induced feeding competition. We observed a behavioral pattern that was consistent both with increased stress and vigilance (self-directed behaviors increased, play decreased, rest decreased) and increased ingroup cohesion (interindividual proximity decreased, aggression over food decreased, and play during feeding competition increased). These results support the hypothesis that outgroup threat elicits ingroup tolerance in chimpanzees. This suggests that in chimpanzees, like humans, competition between groups fosters group cohesion.

## Introduction

The evolution of intergroup relations is of great importance for theories of both human evolution and animal behavioral ecology. Outgroup threat has long been proposed as a key driver of ingroup cohesion and cooperation and has been emphasized as a potentially significant factor in humans' great capacity for both cooperation and aggression [1–4], but to date evidence from our closest relatives in controlled experiments is lacking. Competition between groups over resources may incur a high cost during dangerous aggressive encounters but may also result in territorial expansion enhancing group fitness [2, 5]. Outgroup threat, in particular in situations involving limited resources, has therefore been proposed as a context that may simultaneously select for both greater aggression towards outgroup members and greater

**Funding:** This research was financially supported by the Leading Graduate Program in Primatology and Wildlife Science of Kyoto University, Grant-in-Aid for Scientific Research (JSPS KAKENHI 17H05862 and 19H00629 and MEXT KAKENHI 19H05736 to SY). The funders had no role in study design, data collection and analysis, decision to publish, or preparation of the manuscript.

**Competing interests:** The authors have declared that no competing interests exist.

tolerance towards ingroup members. Under this hypothesis, outgroup threat enhances group cohesion in order to strengthen the group's competitive ability against intruders [6]. Chimpanzees (*Pan troglodytes*) are one of humans' two closest living relatives and engage in both regular intergroup aggression as well as coordinated group-level behaviors such as group hunting [7] and border patrols [8]. Thus, if group level competition has selected for ingroup cohesion and cooperation in humans, proximate cues indicating outgroup threat are predicted to directly stimulate ingroup cohesion in chimpanzees [9].

Some evidence has accumulated that in cooperative breeders with a single breeding pair outgroup conflict solicits ingroup cohesion (e.g. dwarf mongooses (*Helogale parvula*) [10]; green woodhoopoes (*Phoeniculus purpureus*) [11, 12]; cichlid fish (*Neolamprologus pulcher*) [13]), however it is unclear whether this effect generalizes to primates or to species with different social structures. Results from primates on the relation between outgroup competition and ingroup cohesion have so far been mixed. In vervet monkeys (*Chlorocebus pygerythrus*), individuals participating in intergroup encounters subsequently received more grooming after the encounter ended [14]. In contrast, in capuchin monkeys (*Cebus apella*) the presence of outgroup visual aggression from adjacent groups increased the frequency of within group aggression, interpreted as a product of higher general social tension [15]. Similarly, simulated intrusion of outgroup individuals led to increased within group aggression in lion-tailed macaques (*Macaca silenus*) [16] and in the wild Bonnet macaques (*Macaca radiata*) were more aggressive toward their ingroup following intergroup encounters [17]. These results have been interpreted in part to be caused by increased tension following intergroup encounters, negatively affecting within group social relations [15]. Phylogenetic studies on the relation between outgroup competition and ingroup cohesion have similarly been mixed. One review comparing many primate species, including apes, monkeys, and lemurs, found no effect of intergroup aggression on within group affiliation (measured by absolute grooming frequency and territorial overlap) [18] while another across several Old World monkeys and one New World monkey species found an effect only in females (where affiliation was measured by grooming network density) [19].

Among apes, a recent study found that in mountain gorillas (*Gorilla beringei beringei*) affiliative interactions among females increased, agonistic interactions among males decreased, and overall time spent resting decreased immediately following intergroup encounters [20]. A previous study found a decrease in fission-fusion dynamics during border patrols and intergroup encounters in wild chimpanzees, potentially indicative of greater ingroup cohesion [21]. Another recent study on the same population found that party size was larger on days and months with territorial activity (direct outgroup encounters or border patrols) and that frequency of ingroup aggression among males was lower on days with territorial activity [6]. The long-term data suggests a directional component where recent territorial activity leads to decreased group modularity and increased party size in the following days while controlling for major ecological factors, however it remains possible that other factors prompted both behaviors and directionality was not assessed for the short-term effects within a given observation day. On the other hand, captive studies have found that during periods of more frequent vocalizations by neighbouring groups chimpanzees display more frequent self-directed behaviour [22] as well as more frequent intragroup agonism and aggression [23, 24]. However, the neighbouring groups included some former groupmates and were in regular visual and auditory contact with one another, making their group affiliation difficult to interpret and raising the possibility that they just demonstrated contagion of arousal and stress from the neighbouring groups, and thus may respond differently to unfamiliar chimpanzee vocalizations.

To date, most direct research on the influence of outgroup threat on ingroup cohesion in primates has come from field and observational studies. Although these studies have revealed important insights into the evolution of intergroup relations, experiments are necessary to complement these studies and directly test the proximate effects of outgroup threat on within group social behavior in controlled conditions. In humans, several experimental studies have emphasized the role of outgroup competition on group coordination in economic games (e.g. [25–27]). Such experiments with non-human primates are lacking and given mixed evidence from observational studies it remains unclear whether this phenomenon is shared with our closest relatives. Using a playback design, a study with wild chimpanzees demonstrated a clear difference in response to vocalizations of familiar compared to unfamiliar individuals but did not measure the effect on social behavior in detail [28]. Another study with captive chimpanzees additionally found a strong increase in vigilance upon hearing outgroup vocalizations compared to crow vocalizations, which was mediated by level of cortisol release, but again did not measure within group social behavior [29]. Such captive experimental designs allow for both highly controlled experimental designs and opportunities for detailed high-resolution behavioural data from several groups and individuals. No studies have measured the direct impact of outgroup cues on ingroup social behavior in a controlled experimental context with great apes, a necessary piece of data to confirm predictions from theory and field experiments about their relation.

Following previous literature validating its effectiveness in chimpanzees, we used a playback design to simulate the presence of outgroup threat [28–30]. Although group relations in captive environments differ from those of wild chimpanzees, experimental studies have proven effective in other species [15, 16] and playbacks have proven to induce vigilance and stress in chimpanzees [29]. Polizzi di Sorrentino et al. [15] argued that simulated intergroup encounters can be analogized to experimental predator presentation, where despite the captive setting individuals are expected to retain innate behavioral tendencies in response to evolutionarily salient stimuli. We compared chimpanzees' behavior following either outgroup conspecific or control crow vocalizations (following Kutsukake et al. [29]) and then gave semi-monopolizable food to induce feeding competition while recording their behavioral response. If outgroup threat promotes ingroup cohesion, this cohesion may transfer even to times with high within group tension. This would demonstrate the effect is strong enough to withstand additional stress and changes the group's behaviour in feeding contexts among themselves. To date no direct tests in any species have been conducted on how within group feeding competition is influenced by outgroup threat. We thus designed an experimental protocol to examine the effect of outgroup threat on ingroup social behavior and how it is mediated by feeding competition with captive chimpanzees. We had two alternate hypotheses about the effect of outgroup threat on ingroup social behavior.

Hypothesis one (social cohesion hypothesis)—Intergroup competition has selected for an association between outgroup threat and ingroup cohesion in chimpanzees, and thus outgroup stimuli will be a proximate driver of increased ingroup cohesion despite increased stress, even during within group competition over resources.

Hypothesis two (generalized stress hypothesis)—Intergroup competition has not selected for an association between outgroup threat and ingroup cohesion in chimpanzees, and thus outgroup stimuli will increase stress and social tension, decreasing affiliative behaviors and increasing within group aggression, especially during within group competition over resources.

The social cohesion hypothesis therefore predicted that despite increased stress and vigilance from the presence of outgroup threat, individuals would become more affiliative and less aggressive within the group. More specifically, we predicted that rest would decrease while

self-directed behavior would increase, measures for vigilance and stress respectively, that individuals would spend time in closer proximity to one another and that frequency of social grooming and play (as measures of social affiliation and cohesion) would increase, and finally that frequency of aggression would decrease following presentation of outgroup sounds compared to control. The generalized stress hypothesis, by contrast, predicted a general increase in stress and social tension, resulting in a decrease in rest and increase in self-directed behaviors, as in the social cohesion hypothesis, but that inter-individual distance would increase, that grooming and play would decrease, and that aggression would increase in the outgroup compared to control condition. We predicted that these effects would become stronger in a context of tension induced by feeding competition. The social cohesion hypothesis therefore predicted tension would be released through play rather than aggression following playback of outgroup vocalizations, while the generalized stress hypothesis predicted that tension would be released through aggression rather than play. Due to the likelihood of some habituation, we additionally predicted that for some of the effects found there would be an interaction between trial and condition, where the difference between control and outgroup conditions would become smaller across trials.

## Materials and methods

### Ethical note

This research was approved by the ethical committee at Wildlife Research Center, Kyoto University (approval number WRC-2019-KS004A). We carefully considered the ethics of this experiment considering it involved increased stress to the chimpanzees. We emphasize that we investigated behaviours found in natural contexts. These natural stressors are impossible to study without some stress. Still, we note both that we observed increased social cohesion among the chimpanzees and that there were no differences in rates of self-directed behaviour in the food phase between conditions, indicating the induced stress was short lived (see results and discussion for more details). The outgroup playback calls thus likely do not represent a major concern to their welfare.

### Subjects

Subjects were 29 socially-housed adult captive chimpanzees (17 males and 12 females) in five social groups at Kumamoto Sanctuary of Kyoto University, Japan [31]. Chimpanzees were given various environmental enrichment several times per week in addition to daily meals in the morning and evening with additional food spread across their enclosures. All chimpanzees were socially housed with outdoor access and ad libitum access to water and regular health checks. No animals were food or water deprived at any time and no changes were made to their daily schedules beyond playback of sounds and presentation of feeding enrichment. Animal husbandry and research complied with the international standards in accordance with the recommendation of the Weatherall report "The use of non-human primates in research" and all local guidelines. No changes were made to their housing and caretaking following the conclusion of the experiment.

Fifteen males were housed in a simulated fission-fusion grouping structure in three side-by-side enclosures (measuring 128m$^2$, 108.8 m$^2$, and 108.8 m$^2$) with visual access to one another and doors connecting them which can be opened or closed. During all data collection for this experiment doors were closed, forming three social groups of five males each. Group composition was the same on all experimental days in their most common grouping arrangement. The remaining two groups each consisted of one male and either four or eight females (in enclosures measuring 269.5m$^2$ and 150.1m$^2$), which were kept constant throughout the

experiment except for one female who was occasionally housed with another group not involved in this experiment. The majority of the chimpanzees were of the Western chimpanzee subspecies (*Pan troglodytes verus*), though one male in a single male multi female group was an Eastern (*Pan troglodytes schweinfurthii*) and Western chimpanzee hybrid, and another female in the same group was a Nigeria-Cameroon chimpanzee (*Pan troglodytes ellioti*).

## Data collection

Data was recorded and live coded by one observer per group. Each group was observed by the same observer across trials. Observers recorded both scan data on individual behavior and interindividual proximity at two-minute intervals as well as all occurrence data on frequency of aggression and play. Behavioral categories recorded during scans included rest (with posture-either lying down or sitting), social grooming (including giving, receiving, or mutual grooming), self-directed behavior (including both self-grooming and self-scratching), eating, and moving. Proximity at each scan was recorded for each dyad into one of four ordered distance categories including in contact, within arm's reach, <3 meters, and >3 meters (as estimated by the observer) and coded as ordinal data with four levels. The time and individuals involved were recorded for all occurrences of play, aggression (including display, chase, and hit), and copulation. Observers additionally attempted to record instances of vocalizations, but it proved to be too difficult to reliably record all vocalizations and thus was not used in analysis.

## Experimental procedure

Four days of experiments were conducted for each group except for one group for which there were only three days of experiments. Experimental days were separated by at least 3-day intervals to reduce the effect of habituation. Two experimental sessions per day were conducted, once in the morning and once in the afternoon, once with outgroup vocalizations (outgroup condition) and once with crow vocalizations (control condition). The order of which condition was in the morning used an ABBA design across trials, with the order counterbalanced between the all male groups and the single-male multi-female groups. The three all male groups were recorded simultaneously and the two single-male multi-female groups were recorded simultaneously due to being within auditory contact of one another. During each session, observers collected data for 30 minutes before any sounds were played to ensure no abnormal events occurred immediately before the experiments. Data was then collected in the playback phase for approximately 30 minutes while stimuli were played. Caretakers were not always able to deliver the food at precisely 30 minutes after the first sounds played, so the actual time of food delivery was within a range of 28–40 minutes after the first sounds. Analysis was restricted to the minimum duration of time between the first sound and food delivery across sessions. Caretakers then provided two bunches of semi-monopolizable feeding enrichment which is regularly given to the apes and data was recorded for another 30 minutes in the "food" phase. Bundles consisted of carton tubes loosely tied together containing primate chow which can be easily extracted and consumed but cannot be eaten all at once. Tubes could easily be removed from bundles, but each bundle could be held easily by a single individual. Each group received two equally sized bundles at the same time at the start of each food phase, in total including one more tube than the number of individuals in the group. Therefore, the three male groups and the single-male multi-female group of five individuals each received two bundles of three tubes each, and the single-male multi-female group of nine individuals received two bundles of five tubes each.

## Stimuli

Experimental stimuli consisted of pant-hoots of single adult males unfamiliar to any chimpanzees involved in the experiment including some from both wild settings and other captive facilities. Control stimuli consisted of crow "ka" vocalizations. Twelve different high quality recordings of different individuals for each condition were used in the experiment and each was cut or repeated to create a 15 second stimulus for each recording then normalized to consistent volume. Sounds were played at peak pressure level of 95 dB at 10 meters, consistent with the maximum pressure recorded of pant-hoots by Kutsukake et al. [29]. In any given session of experiments, 4 unique recordings were played in order to increase the salience of the stimuli, again following Kutsukake et al. [29]. Recordings were separated by 1 minute of silence, and all 4 were repeated in the same order with the same intervals 15 minutes after the first recording to keep the stimuli salient for the whole observation period. For each of the first three experimental days all recordings played were completely novel, while the recordings on the fourth day were randomly chosen from those used in the first three. Example recordings can be found in S1 File.

## Analysis

All analysis was done using R version 3.5.3 in RStudio [32]. All models reported below were structured in similar ways: As fixed effects, they included condition (outgroup vs. control), trial (1–4, normalized to a mean of 0 and standard deviation of 1), and their interaction. As random effects, they included random slopes of the fixed effects as well as time since start of the phase (for scan behaviors and proximity, normalized) and time of day (morning or afternoon) within individual (nested within group). The proximity data had the same random slopes within dyad, individual 1, and individual 2, where individual 1 and individual 2 were randomly assigned to include both individuals in the dyad. In all models where possible the bobyqa optimizer [33] was used. In the case of non-convergence, we removed random effects in the following way: first, the random slopes for time of day then time since start of phase, then the nesting of individuals within group, then the random slope of the interaction between condition and trial, then the random slope of trial, then the random slope of condition. This sequence was chosen in order to retain the fixed effects whenever possible prioritizing the effect of condition as this was the main hypothesis to be tested in this study, and time since start of phase included more detail and is expected to have had a higher impact than time of day. The same structured simplification was carried out with singular models and the results below present convergent non-singular models, while the maximal singular models are available in S1 File. For the analyses below, we give the formula of the final converging non-singular model. We additionally checked model stability and collinearity of final models; details can be found in S1 File.

Significance was calculated using chi-squared likelihood ratio test with the drop1 function [34] which uses full—null model comparison for hypothesis testing and an alpha value of 0.05. If the interaction between condition and trial was not significant the model was run again with the interaction term removed. This procedure was followed for all variables of interest in separate models for both the playback and food phases. We additionally calculated odds ratio (OR) estimates and 95% confidence intervals for all significant effects. For interactions, the odds ratio represents the odds of a one unit increase when both variables are present over and above the main effects. For example, if there is an interaction between trial and condition the interaction term represents how much the response changes with every trial in the outgroup condition over and above the change that is due to trial and condition alone.

**Proximity.**   Proximity data was modeled with a cumulative link mixed model (CLMM) via the function *clmm* from the *ordinal* package [35]. The proximity data was input as an ordinal measure of 1 (contact), 2 (arm's reach), 3 (<3m) or 4 (>3m) for each dyad and the CLMM was run on the full data. The final model formula for the playback phase was: `proximity ~ trial * condition + (condition*trial|dyad) + (condition*trial| id1) + (condition*trial|id2);` and for the food phase was: `proximity ~ trial * condition + (condition*trial|group/dyad) + (condition*-trial|group/id1) + (condition*trial|group/id2).`

**Behavior scans.**   For self-directed behavior, social grooming, and rest, we used generalized linear mixed models (GLMM) with a logit link function, implemented using the function *glmer* from the package lme4 [34]. The models used the full raw data, where for each individual at each scan, their behaviour was coded as either 0 (not engaged in the behaviour) or 1 (engaged in the behaviour). Another GLMM was run on data restricted to rest data with a 1/0 dependent variable for sitting (1) or lying down (0). For all scan behavior models in both phases the final model formula was: `behavior ~ trial * condition + (condition*trial|individual)` with the exception of self-directed behaviors in the food phase for which the final model formula was: `behavior ~ trial * condition + (condition+trial|individual).`

**All occurrence behavior.**   To model the play and aggression data, we also used GLMMs. For each individual, a 1/0 score was given for whether that individual displayed the behavior in question in the observation period. Thus, rather than absolute frequency, the model used the likelihood of individuals displaying the behavior in the whole time window. This was done to ensure one individual playing for several bouts, or aggressing several individuals, was not overrepresented in the data and because it can be difficult to reliably identify when one bout ends and another begins. This resulted in one data point for each individual per phase per condition per trial. Analyses were run on aggression as a whole rather than for each type of aggression due to the relatively small datasets given by separating each type. For all play and aggression models in both phases the final model formula was `behavior ~ trial * condition + (1|individual).`

## Results

### Proximity

In the playback phase the proportional odds of a dyad being observed in more distant categories was lower in the outgroup condition than the control condition across trials, indicating chimpanzees were significantly closer together ($\beta$ = -0.65, SE = 0.20, $\chi^2$ = 10.81, p = 0.0010; OR = 0.52 (95% CI: 0.36, 0.77); Fig 1). There was no change in proximity in the food phase by condition ($\chi^2$ = 0.46, p = 0.50).

### Behavior scans

Across trials, individuals spent significantly more time engaged in self-directed behaviors in the outgroup compared to control condition in the playback phase ($\beta$ = 1.17, SE = 0.44, $\chi^2$ = 4.84, p = 0.028; OR = 3.21 (95% CI: 1.15, 9.89); Fig 2), but not the food phase ($\chi^2$ = 0.11, p = 0.74). There was a significant interaction between condition and trial in social grooming in the playback phase ($\beta$ = -1.44, SE = 0.43, $\chi^2$ = 9.71, p = 0.0018; OR = 0.24 (95% CI: 0.10, 0.55); Fig 2), where in the first trial individuals engaged in more social grooming in the out-group condition than control condition but this effect decreased across trials. There was no effect of condition on social grooming in the food phase ($\chi^2$ = 0.43, p = 0.51). There was additionally a significant interaction between condition and trial in time spent resting in

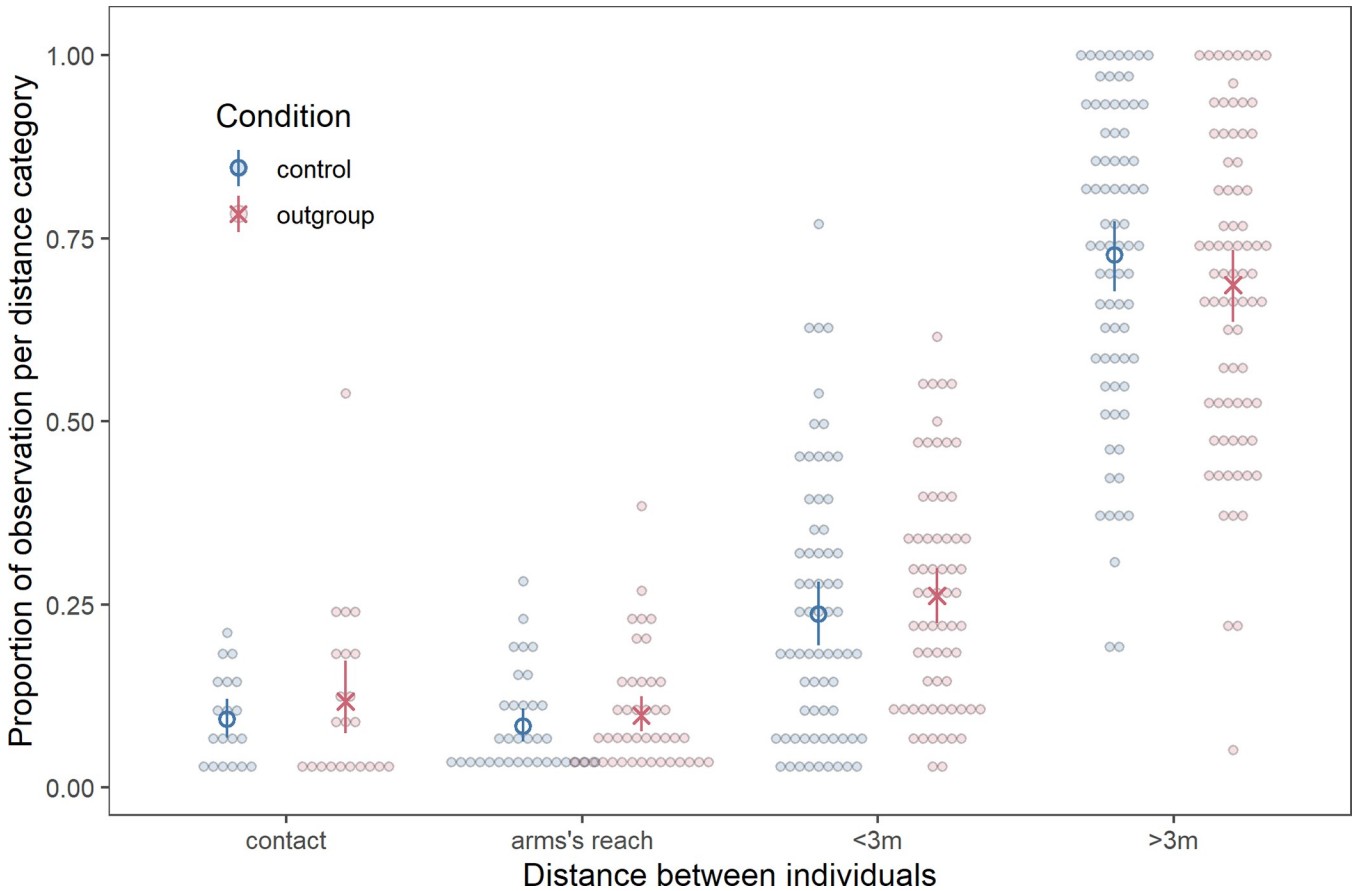

**Fig 1. Proximity in the playback phase.** Proportion of observations in each proximity category in the outgroup and control conditions. Each dot represents a dyad, blue circles represent the mean for the control condition, and red X's represent the mean for the outgroup condition. Red and blue bars represent 95% confidence intervals around the mean (based on a non-parametric bootstrap of the data).

both the playback ($\beta$ = 0.52, SE = 0.15, $\chi^2$ = 9.99, p = 0.0016; OR = 1.69 (95% CI: 1.25, 2.36); Fig 2) and food ($\beta$ = -0.36, SE = 0.11, $\chi^2$ = 8.22, p = 0.0041; OR = 0.70 (95% CI: 0.55, 0.88); Fig 2) phases. In the playback phase, individuals spent less time resting in the first trials in the outgroup than control condition. In the food phase, this pattern was reversed, and individuals spent more time resting in the first trials of the outgroup than control condition. Within rest, in the playback phase there was a significant interaction between condition and trial on posture ($\beta$ = -0.32, SE = 0.14, $\chi^2$ = 4.42, p = 0.036; OR = 0.73 (95% CI: 0.54, 0.98); Fig 2) where individuals in the first trials spent less time lying down in the outgroup compared to control condition but the effect decreased across trials. In the food phase, there was a significant main effect of condition where individuals spent less time lying down in the control condition ($\beta$ = -0.69, SE = 0.21, $\chi^2$ = 9.61, p = 0.0019; OR = 0.50 (95% CI: 0.32, 0.76); Fig 2). Although the sample size was too small to directly investigate the effects of group dynamics (such as sex composition), there was considerable variation between groups. A table of results by group can be found in S1 File.

## All occurrence behaviors

In the playback phase individuals played significantly less in the outgroup condition than the control condition ($\beta$ = -0.89, SE = 0.44, $\chi^2$ = 4.37, p = 0.037; OR = 0.41 (95% CI: 0.16, 0.95); Fig

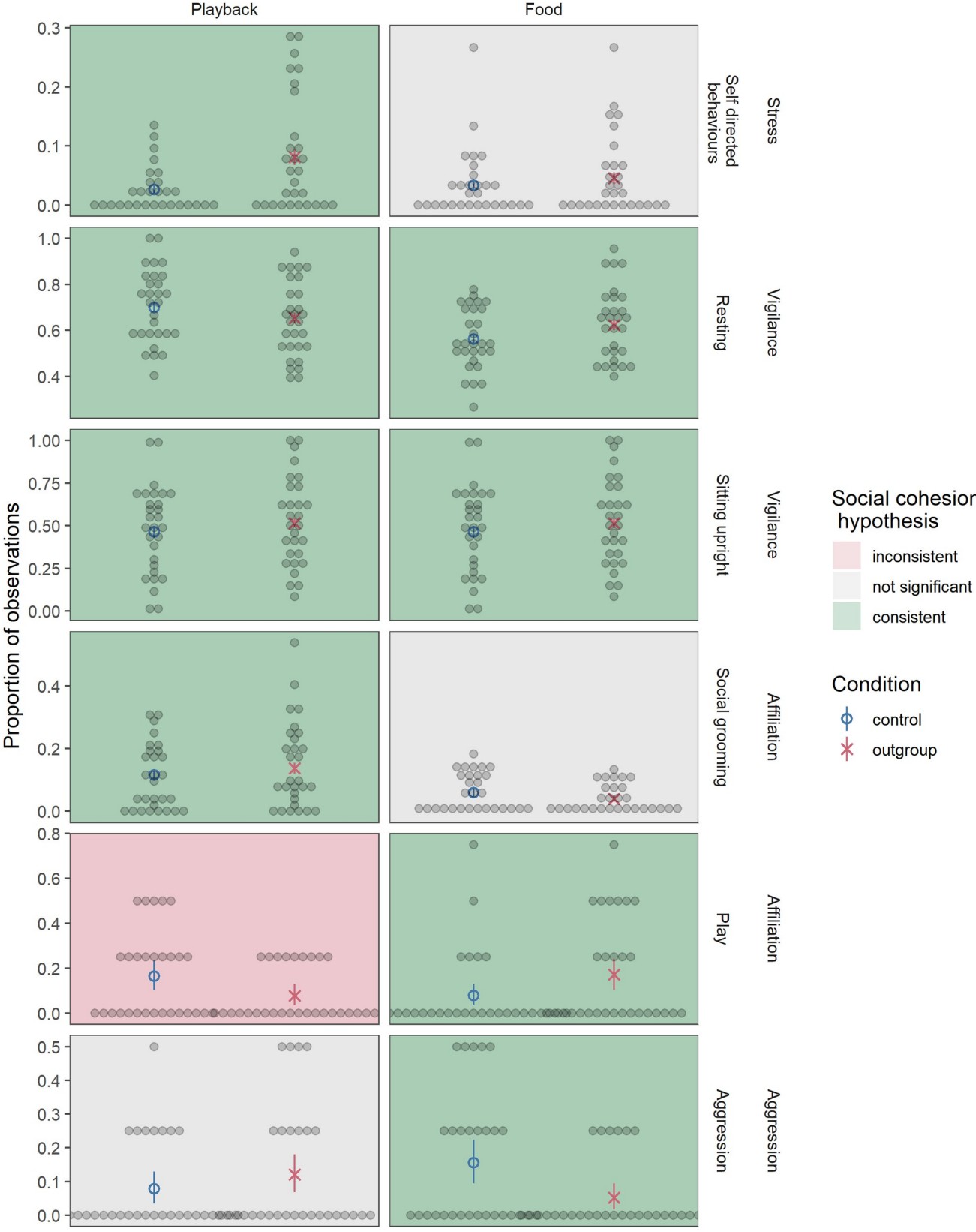

**Fig 2. Behaviors in the playback and food phases.** Social grooming in the playback phase, rest in the playback and food phases, and sitting upright in the playback phase had significant interactions between condition and trial, with the main effect of condition visualized here (graphs depicting trial interactions can be found in S1 File). Each dot represents an individual, blue circles represent the mean for the control condition, and red X's represent the mean for the outgroup condition. Red and blue bars represent 95% confidence intervals around the mean (based on a non-parametric bootstrap of the data).

2). In the food phase, individuals played significantly more in the outgroup condition than the control condition ($\beta$ = 1.15, SE = 0.49, $\chi^2$ = 6.16, p = 0.013; OR = 3.17 (95% CI: 1.27, 8.72); Fig 2). There was no effect of condition on rate of aggression in the playback phase ($\chi^2$ = 1.41, p = 0.24), but the rate of aggression was significantly lower in the outgroup compared to control condition in the food phase ($\beta$ = -1.31, SE = 0.51, $\chi^2$ = 7.47, p = 0.0063; OR = 0.27 (95% CI: 0.09, 0.70); Fig 2). Copulations were rare and statistics could not be calculated.

## Discussion

We observed chimpanzees' behavioral response to outgroup pant-hoots compared to crow vocalizations. Overall, our results were consistent with the social cohesion hypothesis but not with the generalized stress hypothesis. Indicators of stress and vigilance were higher after hearing vocalizations from unfamiliar chimpanzees compared to crow vocalizations but this did not translate into within group tension. Instead, indicators of affiliation and tolerance were higher in the outgroup vocalization condition compared to control crow vocalization condition. Upon receiving semi-monopolizable food, play was higher and aggression lower in the outgroup compared to control condition, indicating a shift towards prosocial strategies in releasing tension induced by feeding competition. These results suggest that outgroup threat directly induces ingroup cohesion in chimpanzees, and importantly, that this effect translates to feeding contexts with high within group tension.

Consistent with previous studies [28, 29], we found behavioral indicators of vigilance and stress increased. More specifically, in the playback phase there were more self-directed behaviors (self-grooming and self-scratching), less rest, and a lower proportion of lying down in the outgroup vocalization condition compared to control crow vocalization condition. For the latter two this effect decreased for later presentations of the vocalizations, presumably due to habituation to the stimuli (see S1 File). The increase in self-directed behaviors, often interpreted as signals of stress [22, 36], is likely due to chimpanzees finding outgroup sounds more stressful than crow vocalizations, consistent with a previous study documenting a rise in cortisol following outgroup auditory stimuli in many of the same individuals as those involved in this study [29]. The decrease in rest, through its interaction with trial, is consistent with field research on gorillas where rest decreased following intergroup encounters [20], but in this case may simply have been due to a trade-off with the relative increase in other behaviors including self-directed behavior and social grooming. The decrease in proportion of rest lying down (as an interaction with trial) may further be interpreted as a sign of vigilance, where chimpanzees remained alert even while not engaged in other behaviors. In the playback phase, contrary to the social cohesion hypothesis, there was a decrease in play in the outgroup compared to the control crow vocalization condition. One explanation is that this was also indicative of increased vigilance or stress. Taken together, the results of several behavioral measures converge on the result that chimpanzees were more stressed and vigilant when outgroup vocalizations were played, compared to crow vocalizations.

Despite the increase in behavioral indicators of stress and vigilance, this stress was not translated into aggression within the group. Much previous research has linked stress levels to aggression through the response of the hypothalamic, pituitary, adrenal (HPA) axis [37–41]. In our study, however, chimpanzees instead engaged in more affiliative behaviors following

the outgroup stimuli compared to the control stimuli. More specifically, individuals moved closer together, indicating that they were spatially more cohesive and tolerant of one another. Further, there was an interaction between condition and trial on social grooming. Chimpanzees engaged in more social grooming in the early trials in the outgroup than control crow vocalization condition, though the difference between conditions decreased across trials, likely due to habituation to the stimuli (see S1 File). These social behaviors indicate that chimpanzees' tolerance, cohesion, and affiliative behavior increased as a result of hearing outgroup vocalizations.

When chimpanzees were given bundles of semi-monopolizable food following the playback of either outgroup or control sounds, there were substantial differences between conditions. Most notably, there was almost no aggression over food observed following the outgroup sounds, whereas there was significantly more aggression over food in the control condition with crow sounds. Further, chimpanzees were more likely to engage in play with one another upon receiving the food after hearing the outgroup vocalizations than crow vocalizations. These results may suggest a change in strategy used to relieve tension. In the control condition, tension led to aggression and competition over food, whereas in the outgroup condition aggression was inhibited and this tension was instead redirected towards play. Captive bonobos (*Pan paniscus*) use play to reduce tension during competition [42] and chimpanzees in this experiment likely used a similar strategy. There was no difference in self-directed behavior, social grooming, or inter-individual distance between conditions in the food phase, and only an interaction between trial and condition in amount of rest. This may be due to the tension release which returned behavior to baseline, though could alternatively be due to rapid habituation to silence or to floor effects due to greater time spent eating and relative reduction in other behaviors. Both total rest and proportion of rest sitting upright were opposite to the effects observed in the playback phase, which may be interpreted as higher feeding competition-induced tension in the control condition, increasing vigilance to the ingroup, while in the outgroup condition there was less tension over food resources and individuals rested and laid down more. The effect of condition on behavior during feeding competition was consistent with the social cohesion hypothesis that outgroup threat promotes within group tolerance, indicating that the prosocial effects of outgroup threat directly transfer to situations of resource competition within the group in chimpanzees.

Although we did not playback ingroup pant hoots (due to constant visual contact between all groupmates, as we did not isolate individuals from their group), we consider it unlikely that the measured results were a product of pant-hoots in general for several reasons. First, previous research have found that familiar pant-hoots (from neighbouring groups with regular visual and auditory contact with one another, including former groupmates) is instead associated with higher rates of ingroup aggression in chimpanzees [24]. This opposite result strongly suggests that pant hoots in general do not cause greater ingroup tolerance and cohesion. Second, all individuals in this study heard pant hoots from familiar individuals in every trial both control and experimental throughout the experiment. It is therefore likely that the identity of the callers, rather than the presence of pant-hoots per se, was the most important difference between conditions. Finally, several measures interacted with trial, indicating that the salience of the vocalizations decreased. This result would not be expected if pant-hoots in general, as opposed to vocalizations of unfamiliar individuals, caused the main effects reported here as all individuals have regularly heard pant-hoots throughout their lives and habituation over the four days of the experiment therefore seems doubtful. For these reasons this study implicates perceived outgroup threat as a driver of ingroup cohesion in chimpanzees as has been demonstrated in humans, though future work testing the effect of familiarity and group membership

will be important to better understand how chimpanzees perceive and respond to other groups.

While the results of this study are promising, there were some important limitations. First, we were not able to directly compare the effect between males and females due to sample size and the lack of any group with both multiple females and multiple males. Field studies with gorillas and chimpanzees have suggested a possible sex difference in response towards outgroup competition [6, 20], but these possible differences could not be studied here. In wild chimpanzees, coalitions of males are most typically the initiators and participants of intergroup encounters [8], so differences might be expected between groups of differing sex composition. In our sample there was considerable variation between groups (see S1 File). It should be noted that in many measures both the all male and single male multi female groups changed in the same direction but unfortunately we do not have sufficient data to test which grouping factors predicted the direction and size of the effect between groups, nor test the effect of condition on individual groups. Still, these are promising future directions for better understanding the impact of outgroup threat on within group social behavior. The significant variation in response suggests behavioral plasticity and the presence of important but variable social dynamics that strongly alter intergroup response. One social group included a female who was not present during all trials and the all male groups are housed in a simulated fission-fusion environment and had constant visual access to one another. Additionally, some caution should be taken about the generalizability of these findings due to the absence of true outgroup threat in daily lives of these captive chimpanzees and the difference between captive and wild feeding competition. Captive experiments on animals' response to outgroup cues have been defended as analogous to experiments on predator responses, where the response to evolutionarily salient stimuli produces similar behavioural effects even in the absence of direct experience [15], but it is worthwhile to note that these chimpanzees have not experienced any true intergroup competition in their adult lives (though roughly half were born in the wild where they may have had some experience of intergroup behaviors as infants). Finally, the form of the stimuli, while allowing for controlled, direct tests, were not fully naturalistic as individual pant hoots spaced at 1 minute intervals is not a typical pattern in the wild [43]. We could not directly test whether behaviour following the first pant hoot and fourth in a given day was different, but chimpanzees in the wild are able to differentiate number of outgroup callers [30]. Testing this effect with varying levels of simulated outgroup threat would therefore be a fascinating future direction. Similarly, the form of feeding competition, where caretakers instantaneously gave bundles of food to groups of chimpanzees, is not the same as the kind of competition that wild chimpanzees would experience, but nonetheless creates a situation with elevated within group feeding competition. Despite these limitations, the results from several different behavioral measures converged on a clear effect of outgroup threat on ingroup cohesion.

The results of this study demonstrate that outgroup threat contributed to ingroup cohesion both in standard grouping dynamics and in a context of feeding competition, but there remain several important unanswered questions for future research. Most notably, future studies should investigate what contributes to the intergroup variation that was observed. As noted, comparisons between groups with differing sex ratios, as well as with differing baseline cohesion, will be essential. It will also be interesting to compare types of food given during the induced feeding competition, especially whether the effects differ between high-value, monopolizable food and low-value, non-monopolizable food. Species level comparisons, especially with bonobos, may additionally reveal the extent to which intergroup competition influences within group tolerance [9]. Bonobos are equally closely related to humans as chimpanzees, and while they do sometimes have aggressive intergroup encounters, unlike chimpanzees they do

not engage in lethal intergroup aggression in the wild [44, 45] and in fact prefer to share food with strangers in captive experiments [46, 47]. Bonobos have been characterized as more tolerant, including both higher performance on dyadic cooperation tasks in captivity [48] as well as lower intensity aggression in the wild [49], but interestingly do not frequently engage in group-level cooperation such as group hunting and border patrols as do chimpanzees [44]. The same experiment with captive bonobos may reveal species level differences in reaction to outgroup stimuli, which would test the hypothesis that chimpanzees' behavior in this paradigm, and the association between human cooperation and competition in experimental contexts, is an evolutionary response to strong intergroup competition. It is also worth noting that the majority of chimpanzees in this study, as the majority of captive chimpanzees generally, are of the Western chimpanzee subspecies. In this subspecies females frequently engage in intergroup encounters, unlike in Eastern chimpanzees (*Pan trogolodytes*) [21], though they display reduced intensity of intergroup encounters overall [50]. Comparisons on the species and subspecies level therefore may prove insightful.

Hormonal mechanisms may further provide a promising future direction. A previous experiment in captive chimpanzees has demonstrated salivary cortisol increases following playback of outgroup vocalizations and correlates with vigilance in response to the stimuli [29], while some human research has found an association between cortisol reactivity and pro-social decision making [51], and thus cortisol release may be a proximate mechanism by which ingroup cohesion is enhanced. Further, oxytocin, a neuropeptide and hormone conserved across mammals, has been strongly implicated in both intergroup behavior and ingroup affiliation in several species [21, 52–58], and may be an important component of the neural and endocrine systems targeted by selection on intergroup behavior [54]. Interestingly, an increase in both cortisol and oxytocin are associated with border patrols in wild chimpanzees, though their release is independent of one another [21, 59]. Future research should measure both baseline oxytocin and cortisol in chimpanzees and bonobos to be compared to one another and to their response in playback experiments. Oxytocin administration in both species, which recently was shown to affect social behaviour differently in bonobos and chimpanzees [60], may further prove to be a promising experimental approach to directly test the hormone's possible role in the evolution of intergroup behaviors. Interindividual, intergroup, and interspecies comparisons of reactions to outgroup stimuli, baseline social behavior, and the underlying hormonal mechanisms may reveal the evolutionary history and mechanisms by which intergroup aggression and ingroup cooperation have evolved in humans.

In sum, we found across several measures, both in the presence and absence of feeding competition, that perceived outgroup threat directly enhances ingroup cohesion and tolerance in captive chimpanzees. This demonstrates that humans' greater group cohesion in competitive contexts is shared with chimpanzees, and suggests that intergroup competition in human evolution may have selected for our ability to maintain cooperation and tolerant relations in large groups in the presence of a common enemy. Several questions remain about the precise evolutionary drivers of this behavioral association and the factors which elicit it, but the results of the current study present strong evidence in a controlled experimental context that it is shared with chimpanzees, and that a comparative approach with great apes may prove a promising direction of study in understanding the evolutionary forces that led to humans' great capacity for group driven behaviors both positive and negative.

## Supporting information

**S1 File. This file contains all supporting information for this manuscript including: Visualizations of effects by trial, table of result by group, stability and collinearity models**

**results, singular model results, all code used in analyses and figure creation (as well as saved models for proximity), all data, both figures, and two examples each of control and outgroup stimuli used in this experiment.**
(ZIP)

## Acknowledgments

We thank Etsuko Nogami and Yusuke Mori, and the other care staff at Kumamoto Sanctuary for their support during and in preparation for the experiments and for their work caring for the chimpanzees. We also thank Drs. Satoshi Hirata, Naruki Morimura, and Fumihiro Kano for their support in this study. We also thank all the chimpanzees at Kumamoto Sanctuary.

## Author Contributions

**Conceptualization:** James Brooks, Shinya Yamamoto.

**Formal analysis:** James Brooks, Manuel Bohn.

**Funding acquisition:** Shinya Yamamoto.

**Investigation:** James Brooks, Ena Onishi, Isabelle R. Clark.

**Methodology:** James Brooks, Manuel Bohn, Shinya Yamamoto.

**Supervision:** Shinya Yamamoto.

**Writing – original draft:** James Brooks.

**Writing – review & editing:** Ena Onishi, Isabelle R. Clark, Manuel Bohn, Shinya Yamamoto.

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
