## [Decision Letter · Decision Letter 0]

26 Nov 2020

PONE-D-20-33650

Uniting against a common enemy: perceived outgroup threat elicits ingroup cohesion in chimpanzees

PLOS ONE

Dear Dr. Brooks,

Thank you for submitting your manuscript to PLOS ONE. After careful consideration, we feel that it has merit but does not fully meet PLOS ONE’s publication criteria as it currently stands. Therefore, we invite you to submit a revised version of the manuscript that addresses the points raised during the review process.

Both reviewers have provided thoughtful and detailed comments on the manuscript. I would like to see you respond to each point they raised, but in particular I think the following points are particularly important to attend to: (i) Clarity of the hypotheses - I agree with R2 that the justification for thinking that feeding competition drives the relationship between outgroup confilict and ingroup cohesion is poor and needs changing or better justifying (ii) Methods - much more clarity on how the data collected resulted in the measures entered into statistical models is required - I could not currently replicate your study with the information you have provided, and as R2 points out its hard to assess the statistical models without clarity as to how the measures that enter them were constructed (iii) inter-group differences - I agree with R1 that you would expect differences in groups of different male composition, so this should be explored in the results (iv) Immediate response to 1st playback - please can you explore in the data if there is a difference in terms of the quality of response to the first playback (where numerical odds are in favour of the multimale groups) vs once they have heard all 4 playbacks, or if your sampling methods do not allow this kind of examination, you could suggest this as an avenue for future research.

We look forward to receiving your revised manuscript.

Kind regards,

Katie E. Slocombe, Ph.D

Academic Editor

PLOS ONE

Journal Requirements:

2. In order to comply with PLOS ONE's guidelines for non-human primate experiments (http://journals.plos.org/plosone/s/submission-guidelines#loc-non-human-primates), please provide additional details regarding housing conditions, feeding regimens, environmental enrichment, and all relevant steps taken to alleviate suffering (anesthesia, analgesia, details about humane endpoints, euthanasia, etc.). Also indicate how often animal care staff monitored the health and well-being of the animals and the criteria used to make such assessments. Lastly, specify the disposition of animals at the end of the study (e.g. euthanasia, returned to home colony, etc.). If animals were euthanized following the study, please provide the method of sacrifice.

Reviewers' comments:

Reviewer's Responses to Questions

**Comments to the Author**

1. Is the manuscript technically sound, and do the data support the conclusions?

Reviewer #1: Partly

Reviewer #2: Yes

2. Has the statistical analysis been performed appropriately and rigorously? 

Reviewer #1: No

Reviewer #2: Yes

3. Have the authors made all data underlying the findings in their manuscript fully available?

Reviewer #1: Yes

Reviewer #2: Yes

4. Is the manuscript presented in an intelligible fashion and written in standard English?

Reviewer #1: Yes

Reviewer #2: Yes

5. Review Comments to the Author

Reviewer #1: This paper presents the results of an experiment to test whether hearing the calls of four unfamiliar conspecifics would group-living chimpanzees to increase behaviors associated with ingroup cohesion and group coordination, or to experience generalized stress.

In its favour, the goal of understanding whether competition between groups tends to increase group cohesion is valuable because this relationship is not well studied; the authors’ introductory review made this point effectively. The experiment had an explicit and well-controlled design that was replicated with 5 groups. Results were analysed with careful statistical models, and were useful in clearly being more consistent with the social cohesion hypothesis. The Discussion was generally appropriate.

Against the paper, two important problems need attention.

First, no effect sizes are provided. Fig. 1 suggests that (although P-values were often low) the differences in magnitude were sometimes relatively small. The authors need to provide some indication of how big the effects were.

Second, I saw no indication that analyses took account of possible differences between groups. This is an important issue because different dynamics are expected between single-male, multi-female groups and all-male groups, given that male coalitions are the active force in intergroup interactions among chimpanzees.

Furthermore, given that the effect sizes appear to be rather small, it is important to report how results varied among groups.

Evaluation.

This study is well designed and presented, but it needs some changes before its significance is fully clear.

Additional comments

Introduction. Most captive chimpanzees are a different subspecies (P.t. verus)from many of the wild chimpanzees for which studies were reported. Since Wilson et al. (2012) found less evidence of intense intergroup aggression in verus, the difference should be mentioned.

L216-226 “Experimental stimuli consisted of pant-hoots of single adult males… In any given session of experiments, 4 unique recordings were played… separated by 1 minute of silence, and all 4 were repeated in the same order with the same intervals 15 minutes after the first recording…”

So in the playback condition, each group heard the calls of 4 different adult males, played back in the same order (with each call separated by 60 seconds of silence). (1) Since this is very artificial, given that free-living chimpanzees tend to produce calls that overlap with each other, the Discussion should briefly acknowledge that the target chimpanzees might have regarded the arrangement as odd. (2) One might expect that the behaviour of the target chimpanzees would change during the playback. After the first call was played, the perception could be that there was an opportunity (for the all-male groups) to conduct an attack. As the number of calls played rose to four, the perceived balance of power would shift in the favour of the playback males. During the first few minutes of the experiment, therefore, one might expect that at least for the all-male groups, the degree of stress would increase. For this reason, I would encourage an analysis that compared responses to the first call with subsequent responses. This is not necessary for the paper to be published, and it may not be possible to do this analysis if there is too little behavioural data in the first minute, but I would at least like to see some acknowledgment of the fact that a single call has a different salience from a collection of calls.

L442 “contributed” (use past tense)

L453 But bonobos do have aggressive intergroup interactions, which should be acknowledged.

L459 Note that these were captive chimpanzees.

Reviewer #2: Review for PONE-D-20-33650

General Comments

This paper experimentally examines the relationship outgroup threat and ingroup cohesion in groups of captive chimpanzees and tests between a social cohesion hypothesis and the alternate generalized stress hypothesis. In the experimental condition, the authors played stranger pant hoots to the chimpanzees and recorded measures of stress, vigilance, affiliation and aggression and compared these behavioral responses to a control condition of playing a crow vocalization. They further examined the behavioral responses to these two conditions after the playback period when the chimpanzees were stressed with a monopolizable food source. The authors found that the chimpanzees increased stress and affiliation in the outgroup condition even during the increased stress of feeding competition providing support for the idea that outgroup threats increase social cohesion. I think this was an interesting experimental paradigm that yielded interesting and compelling results. I have some general comments that I think need to be addressed in a revision followed by some line-by-line comments.

While I think this question is compelling, I found that the authors framed this paper in a confusing way. To me, the social cohesion hypothesis is that chimpanzees when faced with an outgroup threat show increases in within group social cohesion because individuals need to cooperate to defend their territory from this threat. Instead, the authors state the social cohesion hypothesis is that intragroup feeding competition somehow selects for the relationship between outgroup and ingroup competition and cooperation. This sounds a little like an argument from socioecology that if within group feeding competition is too high, then individuals cannot come together to succeed in between group feeding contests. But I don’t think that is the appropriate argument to make here because that has to do with the ability of individuals to even be in a group. Furthermore, I don’t think that the authors are actually testing whether feeding competition selects for the relationship between outgroup threats and ingroup cohesion. Rather I think what is being tested is whether outgroup threat has selected for the relationship between within group affiliation and aggression. This is the comparison between control and the experimental condition in the playback phase. The addition of the feeding competition phase after the playback phase is an additional stressor imposed on the system. I would like to see more explanation in the introduction about why outgroup threats increase group cohesion and think the hypotheses need to be explained better. It would also be helpful to have a better explanation of the predictions in the introduction.

Second, it was a little hard to evaluate the statistical models or really understand the results because there isn’t enough explanation for the metrics that were calculated from the behavioral data. It seems like all of the metrics are some measure of proportion of the scans, but that should be stated explicitly since it might be more appropriate to use the count data with the total number of scans as an offset in these models rather than proportion data. I was also not entirely convinced that there wasn’t a more appropriate way to calculate aggression and play data. This is outlined in more detail below.

Line-by-line comments

Lines 46-50: this seems a bit repetitive with the previous sentences in the introduction.

Lines 56-97: I thought this paragraph could do with some restructuring. First, I think it’s hard to keep track of all of the studies, and maybe it would help to just simplify this without as much detail. I would also have the chimpanzee background be a separate paragraph.

Lines 99-103: While I agree that this question is best investigated experimentally, I am not convinced from these lines that the studies have to be done in captivity, so it’s worth expanding on why that is the case.

Lines 125-130: this is where the idea of the monopolizable food first comes up and it seemed out of place. By introducing it here, it doesn’t seem to have much to do with the stuff in the introduction before this and only starts to fit when you get to the hypotheses. I think the hypotheses should probably be introduced earlier and explained in more detail. But see above about reframing the argument.

Lines 134-142: See above but I disagree that the ingroup feeding competition selects for the relationship between ingroup cohesion and outgroup threat.

Lines 143-157: I think it would be helpful to frame these predictions as stress, vigilance, affiliation and aggression. That way, the authors can specifically describe that self-directed aggression is a proxy for stress, rest is a proxy for vigilance, etc.

Line 146: I think self-directed behavior should decrease.

Lines 164-173: For the 15 males in three groups, how was the composition of these groups determined and were they the same for each trial? Or could the 15 individuals be in different groups from one trial to the other?

Lines 181-183: Are these proximity measures nested? So if individuals are in contact with one another, are they also within 3 meters of one another?

Line 230: Somewhere before you get to the analysis or at the beginning of the analysis section, it would be helpful to know how the data that was collected were turned into metrics that are used in the analysis.

Line 258-262: This is an example of where it would be helpful to know how the data were turned into the metric used here. It’s not totally clear to me whether individual metrics were calculated for each of the four proximity categories.

Lines 273-279: Not clear what the metric is here. Was it some kind of proportion of individuals who exhibited any aggression within the analysis period. I feel like there are a number of ways to calculate aggression including thinking about rates of different kinds of aggression. Displays are quite different than chases and hits which are directed?

Lines 336-338: I’m not sure that this is the place for this but as I was reading this sentence, I thought that a future study or something that could have been done here is to do a control with a non-monopolizable food. That would control for the fact that there might be some excitement or at least an effect of the presence of food.

Lines 356-361: I think it would be helpful in the introduction to explain why play is being used as a measure of social cohesion. It is an affiliative behavior but it’s not one that is often used in metrics of social bonding as is the case for grooming.

Line 441: Also important to note that captive chimps are not facing the same kind of feeding competition and even this kind of simulated feeding competition is not the same.

Line 441: Another factor to consider when thinking about the differences between the groups is the strength of the social bonds within the groups. Some individuals within the groups may have stronger bonds and that might impact certain metrics like grooming, and play.

Figure 2: I think it would be helpful in the text and in this figure to arrange these metrics by categories: stress, vigilance, affiliation and aggression.

6. PLOS authors have the option to publish the peer review history of their article (what does this mean?). If published, this will include your full peer review and any attached files.

Reviewer #1: **Yes: **Richard Wrangham

Reviewer #2: No

---

## [Author Response · Author response to Decision Letter 0]

11 Dec 2020

Dear Professor Slocombe,

Thank you for the opportunity to revise our manuscript. We greatly appreciate the comments made by yourself and both reviewers, all of which were valid and extremely useful in revising our manuscript. Below, we have pasted the reviewers’ responses and detailed our responses. 

Sincerely,

James Brooks, Ena Onishi, Isabelle R. Clark, Manuel Bohn, and Shinya Yamamoto

Reviewer #1: This paper presents the results of an experiment to test whether hearing the calls of four unfamiliar conspecifics would group-living chimpanzees to increase behaviors associated with ingroup cohesion and group coordination, or to experience generalized stress.

In its favour, the goal of understanding whether competition between groups tends to increase group cohesion is valuable because this relationship is not well studied; the authors’ introductory review made this point effectively. The experiment had an explicit and well-controlled design that was replicated with 5 groups. Results were analysed with careful statistical models, and were useful in clearly being more consistent with the social cohesion hypothesis. The Discussion was generally appropriate.

Against the paper, two important problems need attention.

First, no effect sizes are provided. Fig. 1 suggests that (although P-values were often low) the differences in magnitude were sometimes relatively small. The authors need to provide some indication of how big the effects were.

Second, I saw no indication that analyses took account of possible differences between groups. This is an important issue because different dynamics are expected between single-male, multi-female groups and all-male groups, given that male coalitions are the active force in intergroup interactions among chimpanzees.

Furthermore, given that the effect sizes appear to be rather small, it is important to report how results varied among groups.

Thank you for your fair and valuable evaluation of our manuscript. We have considerably revised the paper to account for yours and the other reviewer’s comments and have addressed each point as below. 

Regarding your first major point in need of greater attention, unfortunately there are no standard and agreed upon methods to calculate effect sizes for GLMMs. Still, we recognize the importance of inputting values to allow comparisons of the effect size between studies, and have therefore added odds ratio estimates for each model with significant effects.

Regarding your second major point, we unfortunately do not have the data to fully explore this possibility fairly. While we strongly agree that differences might be expected and this will be an extremely important direction of future study, at this time we are unable to statistically answer with confidence whether such a difference exists. Still, we have expanded considerably on this point in the discussion, both in the limitations (lines 451-461) and future directions (lines 492-493) paragraphs. We additionally add the point that the response did indeed vary quite a bit between groups, there was not an obvious difference in direction between the all male and single male multi female groups, but that this variation will be a worthwhile point of future study (line 458).

Evaluation.

This study is well designed and presented, but it needs some changes before its significance is fully clear.

Additional comments

Introduction. Most captive chimpanzees are a different subspecies (P.t. verus)from many of the wild chimpanzees for which studies were reported. Since Wilson et al. (2012) found less evidence of intense intergroup aggression in verus, the difference should be mentioned.

Thank you for pointing this out, we have made note of this in the methods (lines 188-192) as well as the discussion (lines 508-513), added that in this subspecies females are more likely to participate in intergroup encounters, suggesting subspecies comparisons in addition to species comparisons as a future direction.

L216-226 “Experimental stimuli consisted of pant-hoots of single adult males… In any given session of experiments, 4 unique recordings were played… separated by 1 minute of silence, and all 4 were repeated in the same order with the same intervals 15 minutes after the first recording…”

So in the playback condition, each group heard the calls of 4 different adult males, played back in the same order (with each call separated by 60 seconds of silence). (1) Since this is very artificial, given that free-living chimpanzees tend to produce calls that overlap with each other, the Discussion should briefly acknowledge that the target chimpanzees might have regarded the arrangement as odd. (2) One might expect that the behaviour of the target chimpanzees would change during the playback. After the first call was played, the perception could be that there was an opportunity (for the all-male groups) to conduct an attack. As the number of calls played rose to four, the perceived balance of power would shift in the favour of the playback males. During the first few minutes of the experiment, therefore, one might expect that at least for the all-male groups, the degree of stress would increase. For this reason, I would encourage an analysis that compared responses to the first call with subsequent responses. This is not necessary for the paper to be published, and it may not be possible to do this analysis if there is too little behavioural data in the first minute, but I would at least like to see some acknowledgment of the fact that a single call has a different salience from a collection of calls.

This is an important point we have now added to the manuscript in lines 476-480. We now explicitly mention that the vocalizations’ presentation style is not fully naturalistic, but do provide a test of the hypotheses nonetheless. We also have added your idea as a future direction (lines 480-482) while noting that in this study we do not have enough data to compare changes across the playback phase, this is a fascinating future project. We hope new experiments will be able to examine how the effect observed here relates to assessment of numerical strength of outgroup parties. This is a promising and interesting direction we hope to examine in the future.

L442 “contributed” (use past tense)

Changed, thank you for your attention to detail.

L453 But bonobos do have aggressive intergroup interactions, which should be acknowledged.

We have added this point, thank you for pointing out the need for this clarification (line 501-504).

L459 Note that these were captive chimpanzees.

Added.

Thank you again for your very helpful review of our paper and noting the areas we can improve. We appreciate your advice and thanks to your comments believe our paper is much improved.

Reviewer #2: Review for PONE-D-20-33650

General Comments

This paper experimentally examines the relationship outgroup threat and ingroup cohesion in groups of captive chimpanzees and tests between a social cohesion hypothesis and the alternate generalized stress hypothesis. In the experimental condition, the authors played stranger pant hoots to the chimpanzees and recorded measures of stress, vigilance, affiliation and aggression and compared these behavioral responses to a control condition of playing a crow vocalization. They further examined the behavioral responses to these two conditions after the playback period when the chimpanzees were stressed with a monopolizable food source. The authors found that the chimpanzees increased stress and affiliation in the outgroup condition even during the increased stress of feeding competition providing support for the idea that outgroup threats increase social cohesion. I think this was an interesting experimental paradigm that yielded interesting and compelling results. I have some general comments that I think need to be addressed in a revision followed by some line-by-line comments.

While I think this question is compelling, I found that the authors framed this paper in a confusing way. To me, the social cohesion hypothesis is that chimpanzees when faced with an outgroup threat show increases in within group social cohesion because individuals need to cooperate to defend their territory from this threat. Instead, the authors state the social cohesion hypothesis is that intragroup feeding competition somehow selects for the relationship between outgroup and ingroup competition and cooperation. This sounds a little like an argument from socioecology that if within group feeding competition is too high, then individuals cannot come together to succeed in between group feeding contests. But I don’t think that is the appropriate argument to make here because that has to do with the ability of individuals to even be in a group. Furthermore, I don’t think that the authors are actually testing whether feeding competition selects for the relationship between outgroup threats and ingroup cohesion. Rather I think what is being tested is whether outgroup threat has selected for the relationship between within group affiliation and aggression. This is the comparison between control and the experimental condition in the playback phase. The addition of the feeding competition phase after the playback phase is an additional stressor imposed on the system. I would like to see more explanation in the introduction about why outgroup threats increase group cohesion and think the hypotheses need to be explained better. It would also be helpful to have a better explanation of the predictions in the introduction.

Second, it was a little hard to evaluate the statistical models or really understand the results because there isn’t enough explanation for the metrics that were calculated from the behavioral data. It seems like all of the metrics are some measure of proportion of the scans, but that should be stated explicitly since it might be more appropriate to use the count data with the total number of scans as an offset in these models rather than proportion data. I was also not entirely convinced that there wasn’t a more appropriate way to calculate aggression and play data. This is outlined in more detail below.

Thank you for your review of our manuscript. We fully agree with you on all points, and recognize our manuscript made our main hypothesis unnecessarily confusing in the introduction. As you note, our experiment was aimed to test whether intergroup competition has selected for the relation between ingroup cohesion and perceived outgroup threat. Our food phase was included, as you highlight, to test whether this effect transfers to a time with high ingroup tension where within group affiliation may break down in favour of aggression. We did not mean to imply that ingroup feeding competition was the selection pressure responsible, but that ingroup feeding competition is a prime instance where the effects of outgroup threat may be expected to be prominent and measurable on within group behaviour due to the added social tension. We have changed this discussion to better reflect out motivation and hypotheses (lines 129-132, 138-146, 147-153). We have also briefly expanded on why outgroup threat may increase ingroup cohesion (such as through increasing group defence against intruders) in the introduction as in lines 48-50. 

We have also added more detail to each of our statistical models. More specifically, in the analysis section of the methods, for each type of model we now include sentence clarifying the datasets that were included. We now mention all models were run on raw data. For that reason, offset terms were not necessary as the same number of scans were included in each condition. Regarding the play and aggression data, we have also added more reasoning (lines 302-307) and discuss these in regards to your other points below.

Line-by-line comments

Lines 46-50: this seems a bit repetitive with the previous sentences in the introduction.

We have removed this sentence.

Lines 56-97: I thought this paragraph could do with some restructuring. First, I think it’s hard to keep track of all of the studies, and maybe it would help to just simplify this without as much detail. I would also have the chimpanzee background be a separate paragraph.

While we think it is important to include all of these studies, as the literature on this subject is relatively underdeveloped and all of the previous studies with non-human animals can be described in the space of a few paragraphs, we agree this paragraph was too long and difficult to read. For that reason, we have separated the great ape background into a separate paragraph while retaining the information in the prior paragraph.

Lines 99-103: While I agree that this question is best investigated experimentally, I am not convinced from these lines that the studies have to be done in captivity, so it’s worth expanding on why that is the case.

We have made this sentence less strong (line 101-103) and have added a brief sentence about the benefits of such experiments taking place in captivity (lines 113-115), namely the amount of experimental control and relative ease of consistent high-resolution observational data.

Lines 125-130: this is where the idea of the monopolizable food first comes up and it seemed out of place. By introducing it here, it doesn’t seem to have much to do with the stuff in the introduction before this and only starts to fit when you get to the hypotheses. I think the hypotheses should probably be introduced earlier and explained in more detail. But see above about reframing the argument.

As we mentioned earlier we have rewritten this part to make our point clearer, which we agree was not well stated in the first draft. We instead bring this in as a secondary prediction, where outgroup threat promotes ingroup cohesion, and this then transfers and is especially noticeable in situations with ingroup tension (induced by feeding competition) as in lines 129-132 and 138-146.

Lines 134-142: See above but I disagree that the ingroup feeding competition selects for the relationship between ingroup cohesion and outgroup threat.

Lines 143-157: I think it would be helpful to frame these predictions as stress, vigilance, affiliation and aggression. That way, the authors can specifically describe that self-directed aggression is a proxy for stress, rest is a proxy for vigilance, etc.

Thank you for this suggestion, we agree this makes it easier for readers to quickly understand the predictions from multiple behavioural measures.

Line 146: I think self-directed behavior should decrease.

In both our hypotheses, self-directed behaviour increased, and rest decreased, based on Kutsukake et al.’s (2012) study where cortisol and vigilance rose. While we used different measures, following his study we predicted that stress and vigilance would rise, but that it would be dealt with and its effect on social relations would be different between the conditions.

Lines 164-173: For the 15 males in three groups, how was the composition of these groups determined and were they the same for each trial? Or could the 15 individuals be in different groups from one trial to the other?

We have clarified this point, that we used the same composition on each day, which was simply the grouping composition that is already most frequent for these chimpanzees (lines 184-185).

Lines 181-183: Are these proximity measures nested? So if individuals are in contact with one another, are they also within 3 meters of one another?

Following your main point earlier, we added more detail to the statistical methods, and clarified that the proximity measures were on an ordinal scale of four levels (lines 280-282). A CLMM can then takes raw ordinal data and measures the likelihood of changing from one category to another.

Line 230: Somewhere before you get to the analysis or at the beginning of the analysis section, it would be helpful to know how the data that was collected were turned into metrics that are used in the analysis.

We have added this information at the beginning of the analysis sections for each type of data.

Line 258-262: This is an example of where it would be helpful to know how the data were turned into the metric used here. It’s not totally clear to me whether individual metrics were calculated for each of the four proximity categories.

As mentioned, we used all the proximity data together in a CLMM rather than running analyses on each of the four proximity categories, we made this point clearer in lines 280-282.

Lines 273-279: Not clear what the metric is here. Was it some kind of proportion of individuals who exhibited any aggression within the analysis period. I feel like there are a number of ways to calculate aggression including thinking about rates of different kinds of aggression. Displays are quite different than chases and hits which are directed?

We have added details to this metric, where for each individual in each phase of each condition of each trial, there is a simple 1/0 metric of whether they displayed the given behaviour (lines 290-292, 305-306). This was done because separating bouts was not considered reliable and gives a more robust metric. While we then were not able to compare the different forms of aggression, which we agree are quite different, because of smaller sample sizes and floor effects of any one given form of aggression, we feel confident that these measures are valid and replicable. 

Lines 336-338: I’m not sure that this is the place for this but as I was reading this sentence, I thought that a future study or something that could have been done here is to do a control with a non-monopolizable food. That would control for the fact that there might be some excitement or at least an effect of the presence of food.

Thank you for this suggestion, we agree this is a very interesting question for future research and have added this to the future directions on lines 493-493.

Lines 356-361: I think it would be helpful in the introduction to explain why play is being used as a measure of social cohesion. It is an affiliative behavior but it’s not one that is often used in metrics of social bonding as is the case for grooming.

Relating to your earlier comment, we have clarified how each measure relates to the more general predictions of the group cohesion hypothesis (lines 149-154). We now state more clearly that we take play as another measure of affiliation, and while as you note it is not typically used as a metric of social bonding in the same way, we take it as another useful short term measure of positive social relations because it shows spontaneous affiliative social behaviour with other group members.

Line 441: Also important to note that captive chimps are not facing the same kind of feeding competition and even this kind of simulated feeding competition is not the same.

This is a good point we have added to the paragraph on limitations (lines 482-485).

Line 441: Another factor to consider when thinking about the differences between the groups is the strength of the social bonds within the groups. Some individuals within the groups may have stronger bonds and that might impact certain metrics like grooming, and play.

We strongly agree and think this will be an important and insightful future direction for this line of research. Baseline social relations and their impact on the observed correlation between ingroup cohesion and outgroup threat will be a fascinating future project. We have added this suggestion to lines 492-493.

Figure 2: I think it would be helpful in the text and in this figure to arrange these metrics by categories: stress, vigilance, affiliation and aggression.

Thank you for this suggestion, we have added this to the figure exactly as you suggest and agree in makes the graph more readily understandable by readers at first glance.

Thank you again very much for all your comments and suggestions which we believe have made our manuscript significantly stronger.

---

## [Editor Report · Decision Letter 1]

13 Jan 2021

PONE-D-20-33650R1

Uniting against a common enemy: perceived outgroup threat elicits ingroup cohesion in chimpanzees

PLOS ONE

Dear Dr. Brooks,

Thank you for submitting your manuscript to PLOS ONE. After careful consideration, we feel that it has merit but does not fully meet PLOS ONE’s publication criteria as it currently stands. Therefore, we invite you to submit a revised version of the manuscript that addresses the points raised during the review process.

Thank you for your thorough and thoughtful revision of your manuscript. I think you did an excellent job of addressing the reviewer’s comments and this is shaping up to be a great paper. Careful reading of your revision have yielded a few more areas I would like you to address. All line numbers refer to the track changed version of the manuscript.

Interactions. Whilst your predictions exclusively deal with main effects of condition, many of your models reveal significant interactions. Whilst you are careful to describe the pattern of interaction between trial and condition in the text you don’t provide any illustration of these effects (Figure 2 which is cited represents main effects only).

I would like you to:

produce figures of graphing trial and condition for each of the significant interactions and include them in the supplementary and cite them in the main text;Include some reference to habituation across trials and an interaction with trial being expected in the predictions section of the introduction – so making it clear that either a main effect of condition or an interaction with trial and condition, where the effect of condition decreases with trial would be supportive of the hypothesis (so that Figure 2 can legitimately say the effect is ‘consistent’ with the hypothesis even when it represents an interaction with trial).Be explicit in the figure 2 legend that the measures which produced significant interactions are graphed here as main effects of condition and refer the reader to the new figures in the supplementary if they wish to see the interactions graphed.I am unsure of how to interpret an oddsratio for an interaction – could help the reader to understand what it means?Ethical concerns. Some readers may find the increase in stress the experimental procedure induced ethically questionable. I think it would be good to highlight in the discussion that investigation of a natural event which induces stress (intergroup encounters) is impossible without elevating stress levels, but as there was no difference in stress behaviours in the feeding context, the induced stress effects appeared to be shortlived (30 minutes), and therefore do not represent a substantial welfare issue.In response to R1, you now state in the discussion there was considerable group variation. Please provide some data to demonstrate this in the supplementary (maybe figures showing the effect of condition by group?) and add a short section in the results to introduce this. Group doesn’t make it as a factor in most of your final models, so currently you don’t provide any evidence to support your claim of group variation, and it would be good for the interested reader to be able to look at the variation.

L471-3 I’m not quite clear what the point you are trying to make here – what part of the results could these factors account for (unless its clear what they might explain, saying they are unlikely to explain it doesn’t make much sense!).

L512 – insert ‘they’ between ‘chimpanzees’ and ‘do’.

Finally there are no copy editors for PLOS One, so please proof read your revision carefully before resubmission.

We look forward to receiving your revised manuscript.

Kind regards,

Katie E. Slocombe, Ph.D

Academic Editor

PLOS ONE

---

## [Author Response · Author response to Decision Letter 1]

18 Jan 2021

Dear Professor Slocombe,

Thank you very much for your feedback and the chance to resubmit. We agree will all your suggestions, and have made the relevant changes to the manuscript and have added the requested figures to supplementary information. We additionally revised minor grammar and formatting points, and added a citation to a recently published study validating oxytocin administration affects social behaviour in bonobos and chimpanzees (Brooks et al. 2021). We have pasted your comments below with specific response in blue. Thank you for your consideration of our manuscript.

Sincerely,

James Brooks, Ena Onishi, Isabelle R. Clark, Manuel Bohn, and Shinya Yamamoto

Thank you for your thorough and thoughtful revision of your manuscript. I think you did an excellent job of addressing the reviewer’s comments and this is shaping up to be a great paper. Careful reading of your revision have yielded a few more areas I would like you to address. All line numbers refer to the track changed version of the manuscript.

1. Interactions. Whilst your predictions exclusively deal with main effects of condition, many of your models reveal significant interactions. Whilst you are careful to describe the pattern of interaction between trial and condition in the text you don’t provide any illustration of these effects (Figure 2 which is cited represents main effects only).

This is an important point. We have now made new visualizations for all of the effects for which the interaction was significant in order to clarify for the readers exactly how the effect changed across trials. These can be found in the supporting information zip file.

I would like you to:

i. produce figures of graphing trial and condition for each of the significant interactions and include them in the supplementary and cite them in the main text;

We have added these to the supporting information and refer to them on lines 367-369 400-401, and 425-426.

ii. Include some reference to habituation across trials and an interaction with trial being expected in the predictions section of the introduction – so making it clear that either a main effect of condition or an interaction with trial and condition, where the effect of condition decreases with trial would be supportive of the hypothesis (so that Figure 2 can legitimately say the effect is ‘consistent’ with the hypothesis even when it represents an interaction with trial).

We have added mention of this aspect of our hypotheses on lines 163-166. 

iii. Be explicit in the figure 2 legend that the measures which produced significant interactions are graphed here as main effects of condition and refer the reader to the new figures in the supplementary if they wish to see the interactions graphed.

We have added this information (lines 367-369).

iv. I am unsure of how to interpret an oddsratio for an interaction – could help the reader to understand what it means?

Thank you for pointing this out, odds ratios are less straightforward to interpret with interactions and we have described this on lines 289-293.

2. Ethical concerns. Some readers may find the increase in stress the experimental procedure induced ethically questionable. I think it would be good to highlight in the discussion that investigation of a natural event which induces stress (intergroup encounters) is impossible without elevating stress levels, but as there was no difference in stress behaviours in the feeding context, the induced stress effects appeared to be shortlived (30 minutes), and therefore do not represent a substantial welfare issue.

Thank you for mentioning this point, we have added these details into the methods in the subsection “Ethical note” in order to give readers a clearer understanding of our ethical thinking (lines 171-178). 

3. In response to R1, you now state in the discussion there was considerable group variation. Please provide some data to demonstrate this in the supplementary (maybe figures showing the effect of condition by group?) and add a short section in the results to introduce this. Group doesn’t make it as a factor in most of your final models, so currently you don’t provide any evidence to support your claim of group variation, and it would be good for the interested reader to be able to look at the variation.

Although we do not have sufficient data to include group type as a factor in our models, we recognize the importance of this and have added to the supporting information a table which include values for each group and measure to demonstrate the variation, and added mention of this to the manuscript in lines 360-363 and 479. 

L471-3 I’m not quite clear what the point you are trying to make here – what part of the results could these factors account for (unless its clear what they might explain, saying they are unlikely to explain it doesn’t make much sense!).

We have taken out this sentence. We had no specific predictions regarding how it could account for any part of our results, but mentioned it simply to clarify potential sources of variation. As you note it made the paragraph more confusing and we removed this sentence. Thank you for your suggestion.

L512 – insert ‘they’ between ‘chimpanzees’ and ‘do’.

Done

Thank you again for your careful consideration and advice on our manuscript.

---

## [Editor Report · Decision Letter 2]

28 Jan 2021

Uniting against a common enemy: perceived outgroup threat elicits ingroup cohesion in chimpanzees

PONE-D-20-33650R2

Dear Dr. Brooks,

We’re pleased to inform you that your manuscript has been judged scientifically suitable for publication and will be formally accepted for publication once it meets all outstanding technical requirements.

Kind regards,

Katie E. Slocombe, Ph.D

Academic Editor

PLOS ONE

Additional Editor Comments (optional):

Thank you for carefully addressing all the points I raised - looking forward to seeing this published! Congratulations!
---

## [Editor Report · Acceptance letter]

1 Feb 2021

PONE-D-20-33650R2 

Uniting against a common enemy: perceived outgroup threat elicits ingroup cohesion in chimpanzees 

Dear Dr. Brooks:

I'm pleased to inform you that your manuscript has been deemed suitable for publication in PLOS ONE. Congratulations! Your manuscript is now with our production department. 

Kind regards, 

on behalf of

Dr. Katie E. Slocombe 

Academic Editor

PLOS ONE